# Pyrolyzed or Composted Sewage Sludge Application Induces Short-Term Changes in the Terra Rossa Soil Bacterial and Fungal Communities

Nikola Major [1,*], Jasper Schierstaedt [2], Adam Schikora [3], Igor Palčić [1], Marko Černe [1], Smiljana Goreta Ban [1], Igor Pasković [1], Josipa Perković [1], Zoran Užila [1] and Dean Ban [1,*]

1 Institute of Agriculture and Tourism, Karla Huguesa 8, 52440 Poreč, Croatia
2 Leibniz Institute of Vegetable and Ornamental Crops, Department Plant-Microbe Systems, Theodor-Echtermeyer-Weg 1, 14979 Großbeeren, Germany
3 Julius Kühn-Institut, Federal Research Centre for Cultivated Plants (JKI), Institute for Epidemiology and Pathogen Diagnostics, Messeweg 11/12, 38104 Brunswick, Germany
* Correspondence: nikola@iptpo.hr (N.M.); dean@iptpo.hr (D.B.)

**Abstract:** The addition of compost or biochar to soils is a viable strategy to increase soil organic matter (SOM), especially if the amendments are produced from biomass wastes. The need for sustainable land management without sacrificing agricultural production is critical to alleviate negative impacts on soil quality, including loss of SOM and nutrients. Turning sewage sludge (SS) into compost or biochar can help in lowering its potential negative effects on soil and providing a viable strategy for SS management through its recycling in agriculture. The aim of this study was to evaluate the short-term impact of SS in its composted or biochar form on the fungal and bacterial communities of the Terra Rossa soil by high-throughput sequencing. A greenhouse pot trial was conducted using a 2-factor experiment consisting of amendment type (compost, biochar) and sampling time (Day 0, 30, and 60) as the main factors. The results showed no effect of SS addition on fungal and bacterial species richness, as well as a shift in relative abundance of the fungal phylum Zygomycota and bacterial phylum Firmicutes and Thermomicrobia. Both bacterial and fungal community analyses showed changes when composted sewage sludge was applied. However, only the fungal community differed from the control soil by the end of the 60-day vegetation period of Chinese cabbage.

**Keywords:** sewage sludge; compost; biochar; soil; Terra Rossa; microbiome

## 1. Introduction

Several key factors play a role in soil nutrient availability and fertility among which the organisms living in soil may be the most important [1]. Soil microorganisms are equally important in both natural and managed ecosystems [2]. The composition of soil microorganisms varies depending on soil physicochemical characteristics, with soil pH as the factor with the highest influence [3]. There are thousands of fungal and bacterial taxa present in the soil and their functional characteristics remain largely unknown [4].

Soil quality is a complex matter involving water holding capacity, soil organic matter (SOM) content, cation exchange capacity, and clay content [5]. Due to the increasing pressure on agricultural production through population growth and climate change, soil quality is rapidly deteriorating [6]. The need for sustainable land management without sacrificing agricultural production is critical to alleviate negative impacts on soil quality, including loss of soil organic matter and nutrients [5]. The application of organic fertilizers is seen as a possibility to improve soil physicochemical properties and the microbial community [7,8].

The addition of compost or biochar to soils is a viable strategy to increase SOM, especially if the amendments are produced from biomass wastes. Increased SOM has

multiple benefits on its physical, chemical, and biological traits [5]. Furthermore, the compost application improves the soil's physical properties by decreasing bulk density [9], increasing the soil porosity and plant available water content, and enhancing the stability of soil aggregates [10,11]. Moreover, an increase in nutrient cycling and decrease in soil erosion and acidification were shown under the compost amendment practice [12]. The addition of compost can help in promoting soil microbial activity through the addition of a food supply for heterotrophic microorganisms, optimization of soil physicochemical properties (change in pH, water, air balances, etc.), and the introduction of compost microbes in soil [5]. Organic amendments can cause shifts in the resident soil microbial community, which are likely to be of short duration, depending on the resistance or resilience of the native community [13]. Compared to other organic amendments, biochar is very stable in soil due to its resistance to microbial degradation and its application to soil can be a viable strategy for long-term increase in SOM [5] and decrease in the greenhouse gas emissions, such as $CH_4$ and $N_2O$ [14]. Similar to compost, the biochar amendment decreases bulk soil density and increases field capacity, soil aeration, and aggregate stability [15,16].

Incorporation of organic fertilizers or other soil amendments influences the soil microbial community on different levels [17–19]. On the one hand, the incorporated nutrients would help some organisms, those which are able to efficiently use the newly delivered source of energy and minerals. On the other hand, however, the incorporation of amendments could influence the diversity of soil native community. Moreover, new organisms that are introduced with the fertilizer would be responsible for this influence. The high level of diversity in respect to the microbial community has several advantages. Our results as well as those of other authors suggest that high diversity would benefit the crop plant, for example, via the presence of plant growth-promoting rhizobia (PGPR) or via a suppression of plant pathogens (antagonism) [20,21]. A negative impact of high microbial diversity on the persistence of human pathogens in soil was shown, for example, for *Salmonella enterica* [22]. In this report, the persistence of this bacterial pathogen was longer in field soils with a low level of microbial diversity. Nevertheless, an incorporation of organic amendments into the soil, for example, sewage sludge, can have a positive impact on the persistence of pathogens; it seems that the imported nutrients may help the pathogenic bacteria to be established in this soil environment [17,18].

The results of municipal wastewater treatment are large quantities of sewage sludge (SS), which require environmentally sound processing before final disposal. Moreover, this remains an open and challenging issue for the European Union [23] since the conventional management strategy (e.g., landfilling) is now restricted [24]. Turning SS to compost is a compelling way to bio-transform this material into a product that is no longer classified as waste, lowering its potential toxicity, and relieving sanitation issues [25]. Application of composted SS as a soil amendment provides many advantages in terms of agronomic performance, resource recovery, and environmental protection since it decreases the agricultural use of synthetic mineral fertilizers, facilitates eco-friendly and economically viable SS management, increases soil fertility, and prevents soil degradation [26–29]. On the other hand, the addition of SS biochar to soil can immobilize trace metals and decrease their bioavailability to plants, which is attributed to its high porosity, large surface area, and associated high sorptive capacity [27,30–32]. Furthermore, pyrolyzing SS is among the most-feasible approaches for P recovery [33].

The aim of this study was to evaluate the short-term impact of soil amendment with SS-derived compost or biochar on the fungal and bacterial communities in Terra Rossa soil. Its importance for Croatian agricultural areas is particularly evident since Terra Rossa (red soil) is the major soil type in the largest part of Istria, as well as the karst region all along the Croatian coast and on islands [34]. For Terra Rossa, where various Mediterranean cultures (e.g., olive, grape, fig, citrus, cherry, hazel tree, peach, nectarine, vegetables, and tobacco) are usually cultivated, an intensive fertilization is required with mineral and organic fertilizers due to its low amount of humus and plant-available P [35].

## 2. Materials and Methods

### 2.1. Experimental Setup

A greenhouse pot trial was conducted using a 2-factor experiment consisting of amendment type (compost, biochar) and sampling time (Day 0, 30, and 60) as the main factors. For amending soil with SS-derived compost or biochar, samples of Terra Rossa belonging to the polygenetic cambisols of karst, subtype luvic or typical Terra Rossa [35] were used. The soil was collected according to ISO 18400-104 (2018) [36] at root depth (0 to 30 cm) from the local agricultural area in southern Istria, Croatia (latitude: 45°15′24.00″ N; longitude: 13°54′9.59″ E). The decision on sampling in the rhizosphere zone is associated with its favorable physicochemical characteristics regarding soil fertility due to the presence of root exudates [37]. In this study, Terra Rossa had 0.4% of TOC; 0.9 g/kg d.w. of TN; 0.3 g/kg d.w. of total P; 7.8 g/kg d.w. of total K; pH of 7.5, and electrical conductivity (EC) of 86 μS/cm [38].

Samples of dewatered aerobically stabilized SS were obtained by a local sewage treatment plant, which processes wastewater mainly from domestic, commercial, and agricultural activities [19]. The initial physicochemical characteristics of SS contained 28.5% of total organic carbon (TOC); 39.6 g/kg d.w. of total nitrogen (TN); 26.7 g/kg d.w. of total P; 3.5 g/kg d.w. of total K; pH of 5.8; and EC of 5280 μS/cm [38].

A sample of SS had undergone post-stabilization to produce compost and biochar. The experiment started on 1 June 2017 in a sheltered concrete enclosure (77 × 77 × 77 cm) using the principle of static aerated pile. Composting biomass was prepared by mixing raw SS and carbonaceous structural material, such as wheat straw at a dose of 40 kg of straw per 1 m³ of SS to adjust the C/N and promote the aeration of initial mixture. Temperature evolution in the middle of the compost was measured daily using the HOBO temperature sensors connected to HOBO Data Logger (U12 4-External Channel). The aeration was enabled monthly by hand turning the pile into the empty bays of their respective enclosures with shovels. The moisture content was maintained between 55% and 60% by manual irrigation. The composting process lasted about 3 months and was completed according to compost maturity and quality test schemes [39,40]. At maturity, the SS compost contained 28.7% of TOC; 34.1 g/kg d.w. of TN; 23.5 g/kg d.w. of total P; 5.1 g/kg d.w. of total K; pH of 6.2; and EC of 3210 μS/cm [38].

Furthermore, the biochar was produced using a Kon-Tiki system where the concept of a flame curtain kiln for biomass pyrolyzation was employed [41]. The process, i.e., temperature in the main pyrolysis zone (from 410 to 470 °C) was monitored using a NiCrNi thermoelement (NiCrNi thermoelement, Elektron Erma-Strmec d.o.o, Stubičke Toplice, Croatia). The resultant biochar contained 23.2% of TOC; 34.4 g/kg d.w. of TN; 41.9 g/kg d.w. of total P; 5.1 g/kg d.w. of total K; pH of 7.4; and EC of 1810 μS/cm [38].

Samples of substrates were produced by the addition of composted or pyrolyzed SS to undisturbed Terra Rossa soil according to the Croatian legislation on sewage sludge agricultural use (NN 38/08) by considering P-fertilization requirements for Chinese cabbage cultivation, where a 10-times higher dose (120 mg/L of P) was only used as no effect was expected at the lower legislative-based recommendation (12 mg/L of P). For this reason, 17.5 and 9.8 g of compost or biochar, respectively were added to each pot (3 L) containing around 3800 g of Terra Rossa. The P-based application was considered since Terra Rossa is insufficiently supplied by plant-available P, significantly below the critical threshold needed for most crops [35]. Compost and biochar were sieved (2 mm) prior to mixing to assure their even distribution within the plant pot. The control treatment represents the unamended substrate.

The pot trial was set under greenhouse conditions with two treatments and one control. Pots were arranged in a randomized position in four replicates. The greenhouse experiment started on 13 September 2017 (Day 0), when the transplants of a hybrid cultivar (PREDURO F1, Takii Europe B.V., St. Annaland, The Netherlands) of Chinese cabbage (*Brassica rapa* L. subsp. pekinensis (Lour.) Hanelt) were planted into plastic pots (3 L) filled with soil amended by SS compost and/or biochar. During the experiment, the standard agronomic

measures for growing Chinese cabbage (e.g., irrigation, fertilization, plant protection, etc.) were performed [20]. Soil was sampled on planting day (Day 0) and again on Day 30 and 60 of Chinese cabbage vegetation.

### 2.2. DNA Extraction and Sequencing

Total community DNA was extracted from 0.5 g of the obtained oil samples, using the FastDNA SPIN Kit for Soil (MP Biomedicals, Heidelberg, Germany) and purified with GENECLEAN SPIN Kit (MP Biomedicals, Heidelberg, Germany) according to the manufacturer's instructions. DNA concentration and purity were monitored on 1% agarose gels. According to the concentration, DNA was diluted to 1 ng/µL using sterile water. Amplicons of 16S rRNA and ITS genes of distinct regions (16SV4-V5 and ITS2) were amplified using the following primers 16S V4-V5: 515F-907R [42] and ITS2: ITS3-ITS4 [43] with the barcode. All PCR reactions were carried out with Phusion® High-Fidelity PCR Master Mix (New England Biolabs, Ipswich, MA, USA). The quality of the PCR products was checked by electrophoresis on 2% agarose gels. PCR products were mixed in equidensity ratios and purified with Qiagen Gel Extraction Kit (Qiagen, Hilden, Germany).

Sequencing libraries were generated using NEBNext® Ultra DNA Library Prep Kit for Illumina following the manufacturer's recommendations and the index codes were added. The library quality was assessed on the Qubit@ 2.0 Fluorometer (Thermo Scientific, Waltham, MA, USA) and Agilent Bioanalyzer 2100 system. Finally, the library was sequenced on an Illumina platform and 250 bp paired-end reads were generated. The raw sequencing data were deposited at the NCBI Sequence Read Archive database, under the accession number PRJNA871992.

### 2.3. Data Analysis

Paired-end reads were assigned to samples based on their unique barcode and truncated by cutting off the barcode and primer sequence. Paired-end reads were merged using FLASH (V1.2.7, http://ccb.jhu.edu/software/FLASH/, accessed on 11 September 2019) [44], a very fast and accurate analysis tool, which was designed to merge paired-end reads when at least some of the reads overlap the read generated from the opposite end of the same DNA fragment, and the splicing sequences were called raw tags. Quality filtering on the raw tags was performed under specific filtering conditions to obtain the high-quality clean tags [45] according to the QIIME (V1.7.0) quality controlled process [46]. The tags were compared with the reference database (Gold database, http://drive5.com/uchime/uchime_download.html, accessed on 11 September 2019) using UCHIME algorithm (UCHIME Algorithm) [47] to detect chimera sequences, and then the chimera sequences were removed [48].

Sequence analysis was performed by Uparse software (Uparse v7.0.1001) [49]. Sequences with ≥97% similarity were assigned to the same OTUs. Representative sequence for each OTU was screened for further annotation. For each representative sequence, the GreenGene Database [50] was used based on RDP 3 classifier (version 2.2, http://sourceforge.net/projects/rdp-classifier/, accessed on 11 September 2019) to annotate taxonomic information. OTUs abundance information were normalized using a standard of sequence number corresponding to the sample with the least sequences. Subsequent analyses of alpha and beta diversity were all performed based on the output normalized data.

Univariate statistical analyses were performed using the Statistica software package version 13.4 (Tibco Software Inc., Palo Alto, CA, USA). Species richness, NMDS, PERMANOVA, and ANOSIM were calculated with the MicrobiomeAnalystR package [51] and plotted using Microsoft Excel.

Differential abundance analysis was performed with ANOVA with centered log-ratio normalized counts for fungal counts as well as limma [52] with additive log-ratio normalized counts after selecting the method using DAtest [53].

## 3. Results

### 3.1. Species Richness and Community Analysis

A total of 4013 fungal OTUs were obtained from the sequencing run with an average count of 74171 reads per sample. After rarefaction, the normalized fungal species richness ranged from 119 to 319 species per sample (Figure 1). The average observed species richness at Day 0 was 268, at Day 30, 248, and at Day 60, 210 OTUs. The average observed species richness in the SS biochar addition treatment was 214, in composted SS addition 232, and in the soil without amendment (control) 249. ANOVA did not show any significant differences between the observed timepoints and biochar/compost addition or the interaction between the main factors.

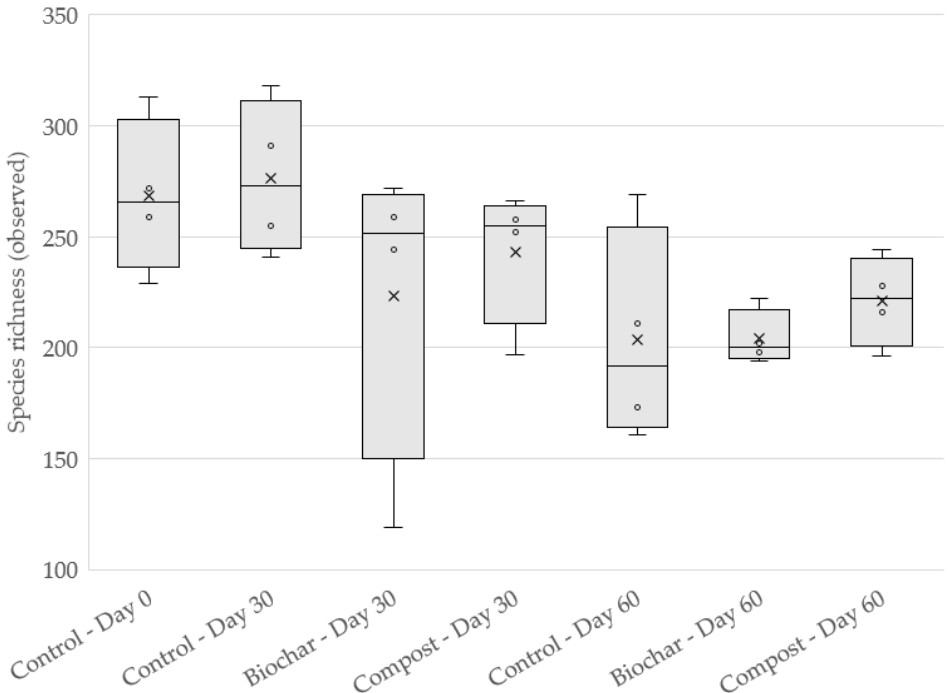

**Figure 1.** Fungal species richness by time and sewage sludge processing method. Level of significance: $p$ (time) = 0.054; $p$ (processing method) = 0.226; $p$ (time $\times$ processing method) = 0.946.

A total of 7324 bacterial OTUs were obtained after the sequencing run with an average of 47,136 reads per sample. After rarefaction, bacterial richness varied from 1072 to 1265 observed species per sample (Figure 2). The species richness differed significantly per timepoint with average values of 1255 at Day 0, 1220 at Day 30, and 1195 at Day 60. The difference between composted SS and SS biochar or soil without amendments was not significant with average values of 1218, 1216, and 1210, respectively. The interaction between the main factors was also not significant.

The PERMANOVA revealed significant differences in the species community between timepoints and the SS processing method as well as the interaction between the main effects (Figure 3). Subsequently, NMDS analysis showed that the biggest difference in fungal species community occurred between Day 30 and 60 with Day 0 being closely positioned with Day 30 (Figure 3). Moreover, the NMDS analysis showed that there is a shift in the fungal community with the addition of the composted SS compared to SS biochar addition or control soil (Figure 3). ANOSIM confirmed the results obtained from PERMANOVA, where significant differences were observed between timepoints ($p < 0.001$), processing method ($p < 0.031$), and the interaction between the main effects ($p < 0.001$).

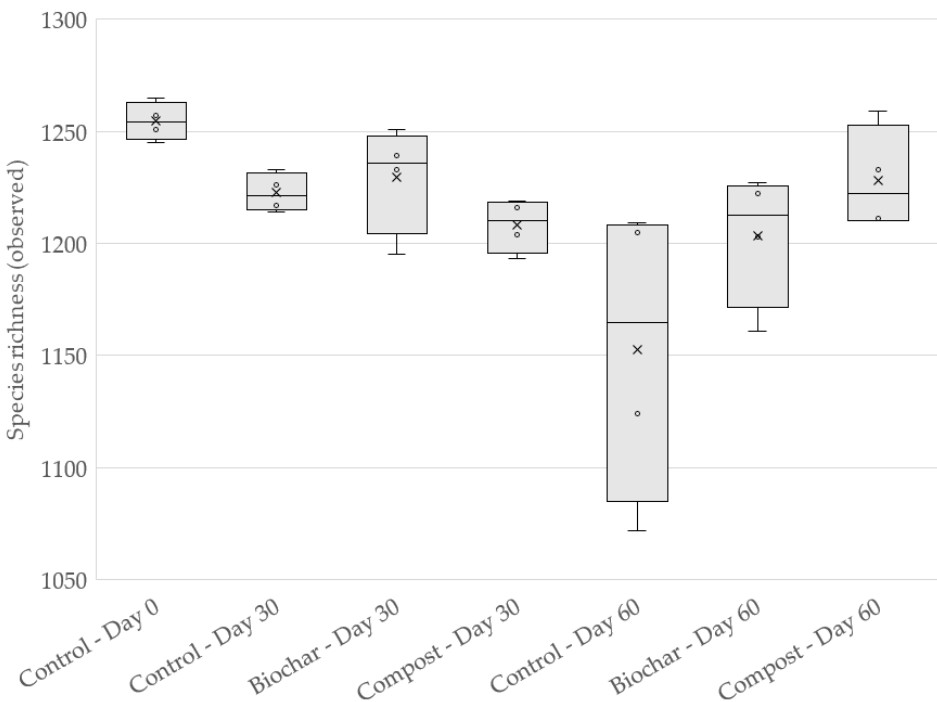

**Figure 2.** Bacterial species richness by the time and sewage sludge processing method. Level of significance: $p$ (time) = 0.002; $p$ (processing method) = 0.896; $p$ (time × processing method) = 0.148.

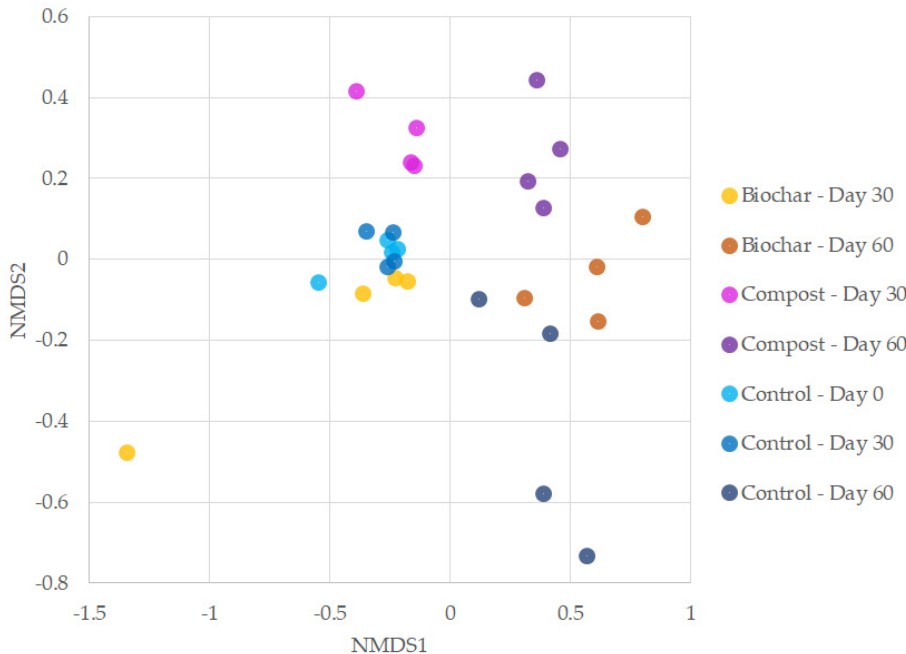

**Figure 3.** Fungal species community analysis by time and sewage sludge processing method. Level of significance by PERMANOVA: $p$ (time) < 0.001; $p$ (processing method) = 0.046; $p$ (time × processing method) < 0.001. NMDS stress = 0.10682.

Bacterial species communities differed significantly between timepoints, but not between the SS processing methods, according to PERMANOVA (Figure 4). However, the interaction between the timepoints and processing methods was significant (Figure 4). The biggest shift in bacterial communities occurred between Day 30 and 60 with Day 0 being close to Day 30 (Figure 4). The difference between the SS processing methods was not evident on Day 60, but on Day 30 there was a noticeable shift in the bacterial community with

the addition of composted SS compared to SS biochar or control soil (Figure 4). ANOSIM confirmed the results of PERMANOVA with significant differences between timepoints ($p < 0.001$) and the interaction between timepoints and SS processing methods ($p < 0.001$). There were no significant differences between the processing methods as the main factor ($p < 0.147$).

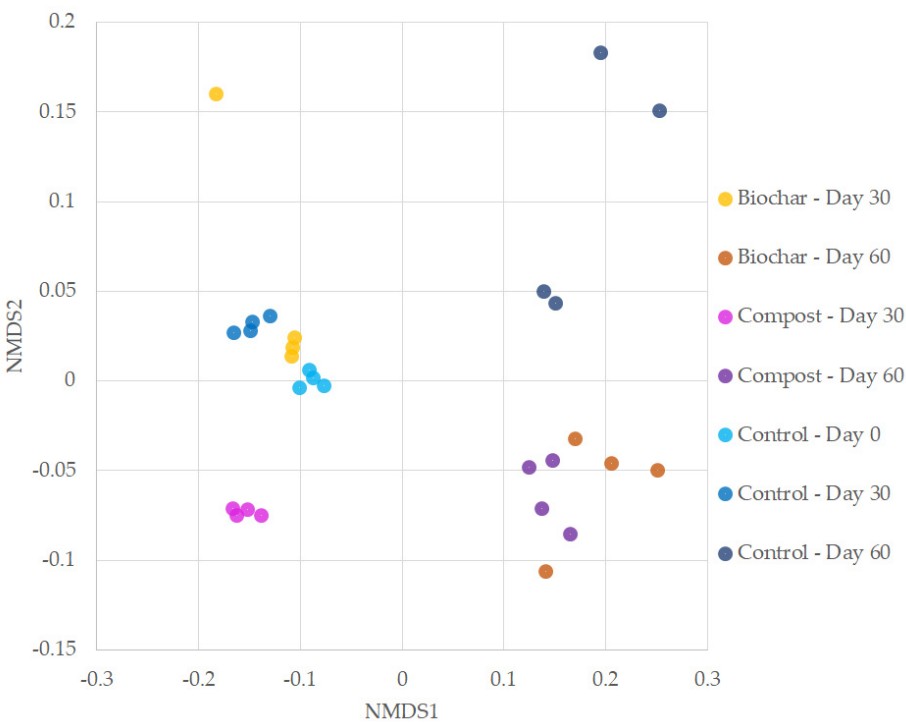

**Figure 4.** Bacterial species community analysis by time and sewage sludge processing method. Level of significance by PERMANOVA: $p$ (time) < 0.001; $p$ (processing method) = 0.169; $p$ (time $\times$ processing method) < 0.001. NMDS stress = 0.063875.

### 3.2. Taxonomic Classification

After referencing fungal OTUs against a database, four major phyla and eleven classes were obtained (Figure 5a,b). Of the four phyla, only Basidiomycota were less than 1% abundant and 27% to 60% of the OTUs were assigned as unknown or uncultured (Figure 5a). For investigative purposes, the OTUs were classified according to the SS processing method of the soil amendment (Figure 5). The most abundant known phylum in all samples was Ascomycota with 40.2%, 17.5%, and 49.9% mean relative abundance for composted SS, SS biochar, and control soil, respectively (Figure 5a). The mean relative fungal phyla composition in soil with composted SS amendment was Ascomycota 40.2%, Chytridiomycota 5.0%, Zygomycota 1.6%, and Basidiomycota 0.2% (Figure 5a). In soil with SS biochar amendment, the mean relative fungal composition was Ascomycota 17.5%, Chytridiomycota 15.3%, Zygomycota 6.9%, and Basidimycota 0.3% (Figure 5a). For control soil, the relative fungal abundance was Ascomycota 49.9%, Chytridiomycota 7.6%, Zygomycota 14.6%, and Basidiomycota 0.9% (Figure 5a). ANOVA revealed significant differences between Zygomycota relative abundances showing a decrease in this phylum in soils with SS biochar amendment and even further in soils with composted SS ($p < 0.011$). Other differences were not significant probably due to high variations in relative abundances between timepoints (data not shown).

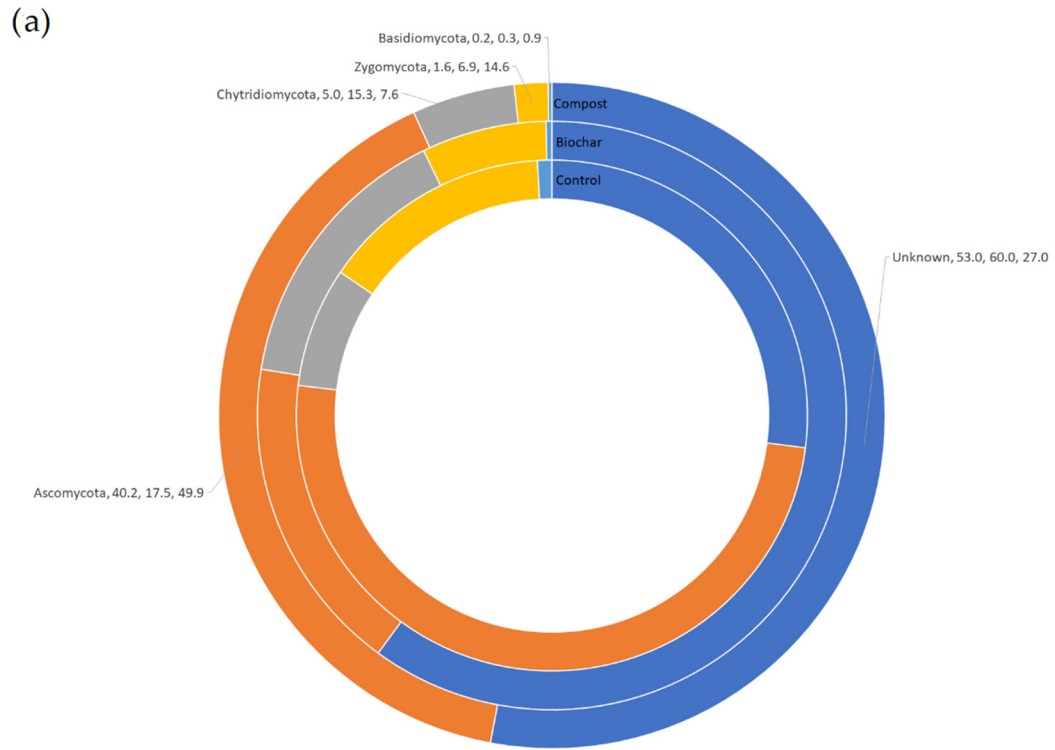

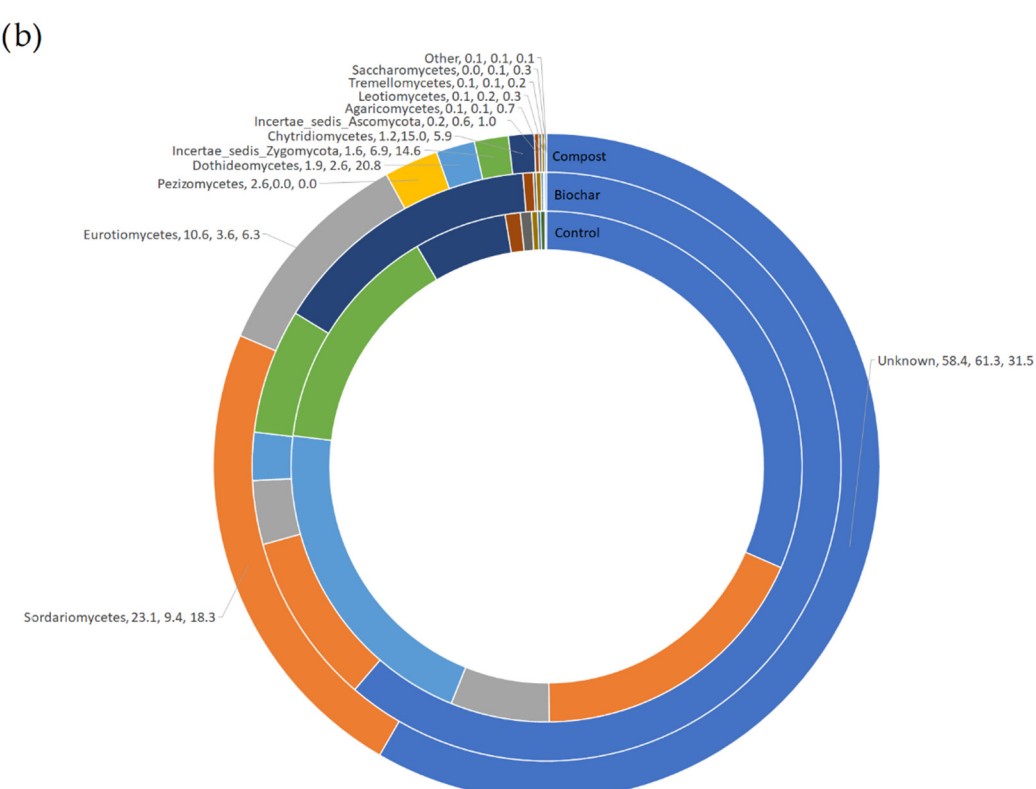

**Figure 5.** Relative abundances of fungal phyla (**a**) and classes (**b**) in Terra Rossa soil after sewage sludge compost or biochar application compared to control soil.

On the class level, unknown or uncultured OTUs were found in relative abundance of 31.5% in control soil, 58.4% in soil amended with composted SS, and 61.3% in soil with SS

biochar amendment (Figure 5b). The most abundant class in soil with composted SS was Sordariomycetes with 23.1%, followed by Eurotiomycetes (10.6%) and Pezizomycetes (2.6%) (Figure 5b). In soil with SS biochar amendment, the most abundant classes were Chytridiomycetes (15.0%), Sordariomycetes (9.4%), unknown Zygomycota (6.9%), Eurotiomycetes (3.6%), and Dothidemycetes (2.6%) (Figure 5b). In control soil, the most abundant classes were Dothidemycetes (20.8%), Sordariomycetes (18.3%), unknown Zygmycota (14.6%), Eurotiomycetes (6.3%), and Chytridiomycetes (5.9%) (Figure 5b). Significant differences were observed for Pezizomycetes with relative abundance of 2.6%, compared to none in SS biochar amended soil or control soil ($p < 0.030$) (Figure 5b). Unknown Zygomycota decreased significantly ($p < 0.014$) in relative abundance in soils with SS biochar amendment (6.9%) and composted SS (1.6%), compared to control soil (14.6%) (Figure 5b). On the other hand, Chytridiomycetes significantly increased ($p < 0.015$) in relative abundance in soils with SS biochar addition (15.0%) and decreased in soils with composted SS amendment (1.2%), if compared to control soil (5.9%) (Figure 5b).

Bacterial OTUs were assigned to 17 phyla with relative abundances over 0.1%, which represented 99.8% of all sequences (Figure 6a). The major phyla in soil with composted SS, soil with SS biochar, and control soil were Proteobacteria (47.2%, 47.6%, and 45.0%, respectively), Actinobacteria (30.2%, 29.8%, and 32.1%, respectively), Chloroflexi (4.2%, 4.1%, and 4.4%, respectively), Acidobacteria (3.9%, 4.8%, and 4.6%, respectively), Bacteriodetes (3.5%, 4.4%, and 4.5%, respectively), Firmicutes (2.2%, 1.3%, and 1.2%, respectively), Nitrospirae (2.0%, 1.8%, and 0.9%, respectively), Thermomicrobia (1.6%, 0.8%, and 0.9%, respectively), Cyanobacteria (1.4%, 1.2%, and 1.7%, respectively), Gemmatimonadetes (1.2%, 1.3%, and 1.4%, respectively), and Verrucomicrobia (0.9%, 1.1%, and 0.9%, respectively) (Figure 6a). Other bacterial phyla had less than 1% relative abundance in all soil samples. Significantly higher relative abundance of Firmicutes ($p < 0.011$) was observed in soil with composted SS amendment (2.2%) compared to soil with SS biochar amendment (1.3%) and control soil (1.2%) (Figure 6a). Moreover, significantly higher Thermomicrobia relative abundance ($p < 0.001$) was detected in soil with composted SS amendment (1.6%) compared to soil with SS biochar (0.8%) amendment or control soil (0.9%) (Figure 6a).

On the class level, the 27 most abundant classes are shown in Figure 6b. The most abundant Proteobacteria in all samples were Betaproteobacteria and Alphaproteobacteria (Figure 6b). Significantly higher relative abundance of Bacilli ($p < 0.028$) was observed in soil with composted SS amendment (2.1%) compared to soil with SS biochar amendment (1.1%) and control soil (1.1%) (Figure 6b). Moreover, significant decrease in relative abundance was observed in KD4 96 class ($p < 0.018$) and Acidimicrobiia ($p < 0.045$) in both composted SS (0.9% and 1.0%, respectively) and SS biochar (0.9% and 1.0%, respectively) amended soil compared to control soil (1.3% and 1.2%, respectively) (Figure 6b).

### 3.3. Differential Abundant Taxa

The ANOVA using centered log-ratio normalized fungal counts revealed 44 taxa that are differentially abundant between processing methods and days (Figure 7). Most of the significant different taxa are more abundant in samples of only one specific processing method and day. However, some of the taxa were more abundant in several samples at the same day, e.g., Family Ascobolaceae or Family Pyronemataceae were more abundant at Day 30 or Processing method, e.g., Metarhizium, or Class Agaricomycetes were more abundant in control samples.

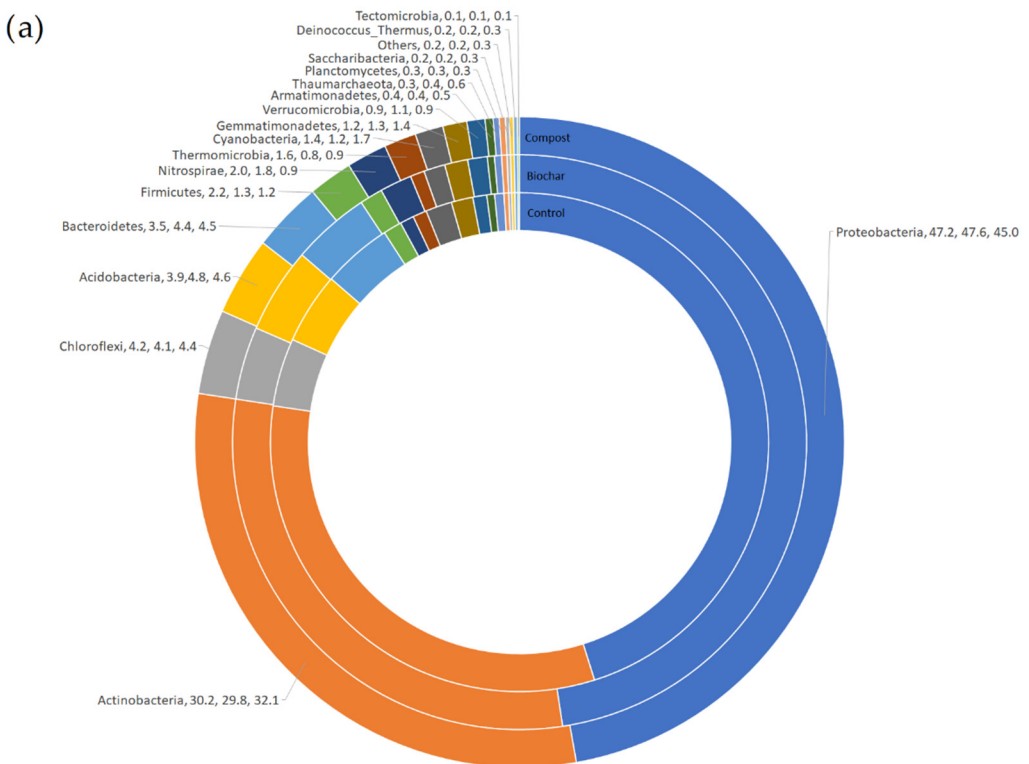

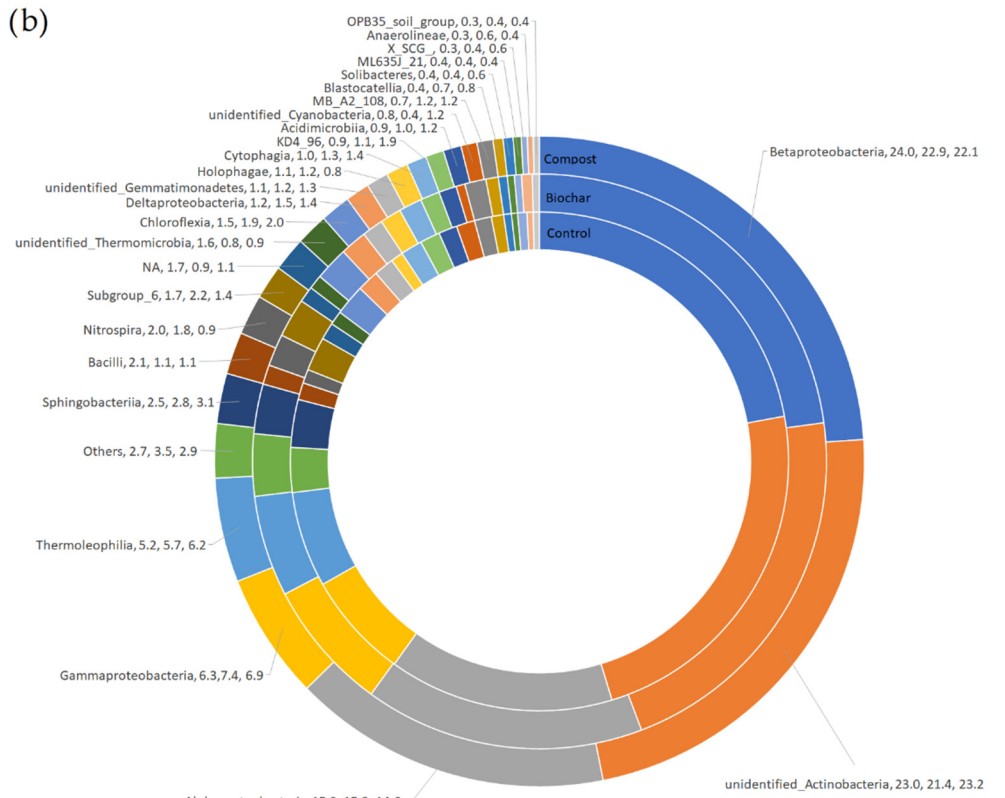

**Figure 6.** Relative abundances of bacterial phyla (**a**) and classes (**b**) in Terra Rossa soil after sewage sludge compost or biochar application compared to control soil.

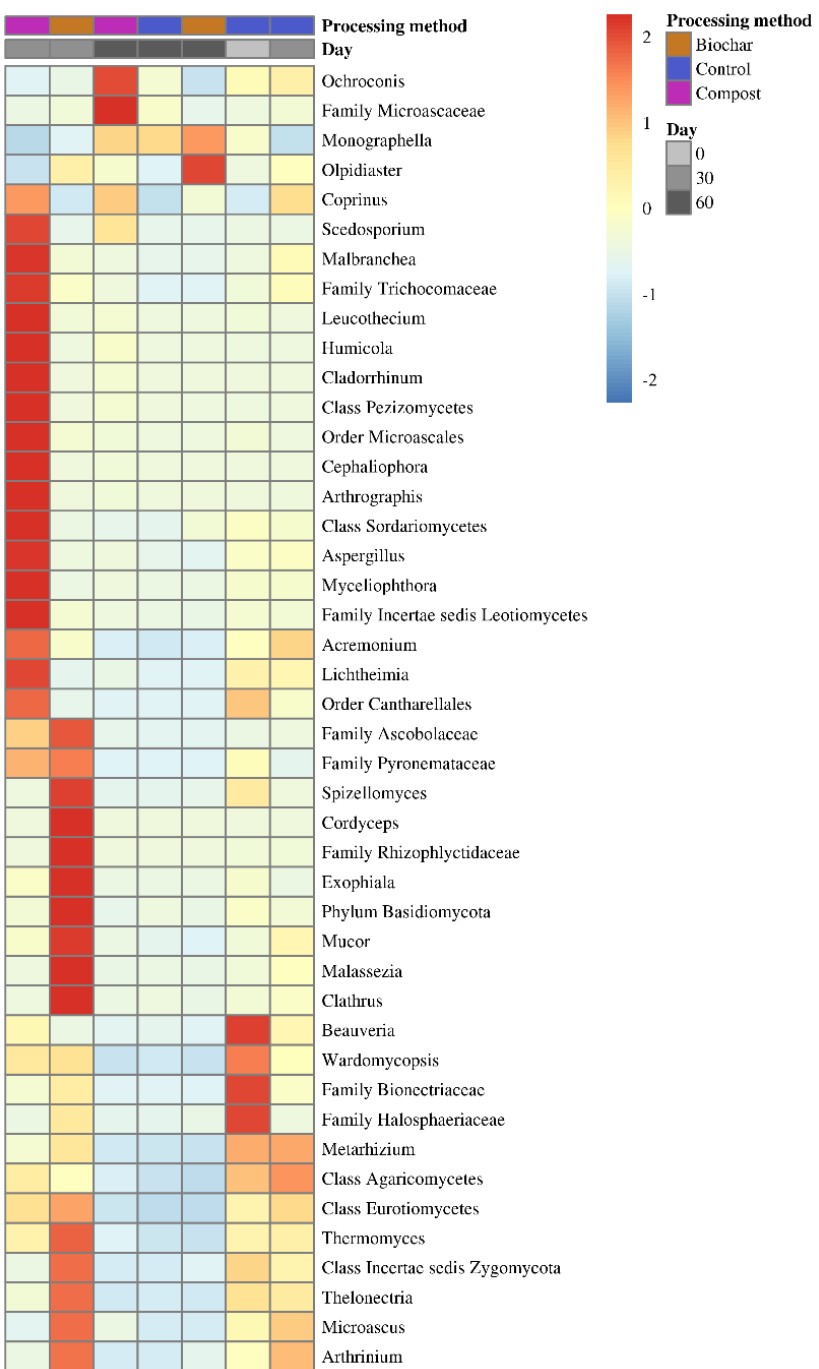

**Figure 7.** Fungal taxonomic groups, which significantly differ in their abundance between processing methods and days, were identified using ANOVA with centered log-ratio normalized counts. A *p*-value of <0.05 was considered as significant. The individual taxa are represented using a colour-code (center-scaled mean of read counts out of four replicates).

For bacteria, 220 taxa were identified using limma with additive log-ratio. The 50 most abundant taxa are shown in Figure 8. Most of the taxa showed clear patterns according to the days. However, some taxa were found to be differentially abundant between processing methods without considering the day as a covariate (purple boxes). These taxa, Rhodococcus, Actinomadura, Order AKYG1722, Family Micromonosporaceae, Phylum Chloroflexi, and Bacillus seem to be present in higher abundance in compost samples if compared to others.

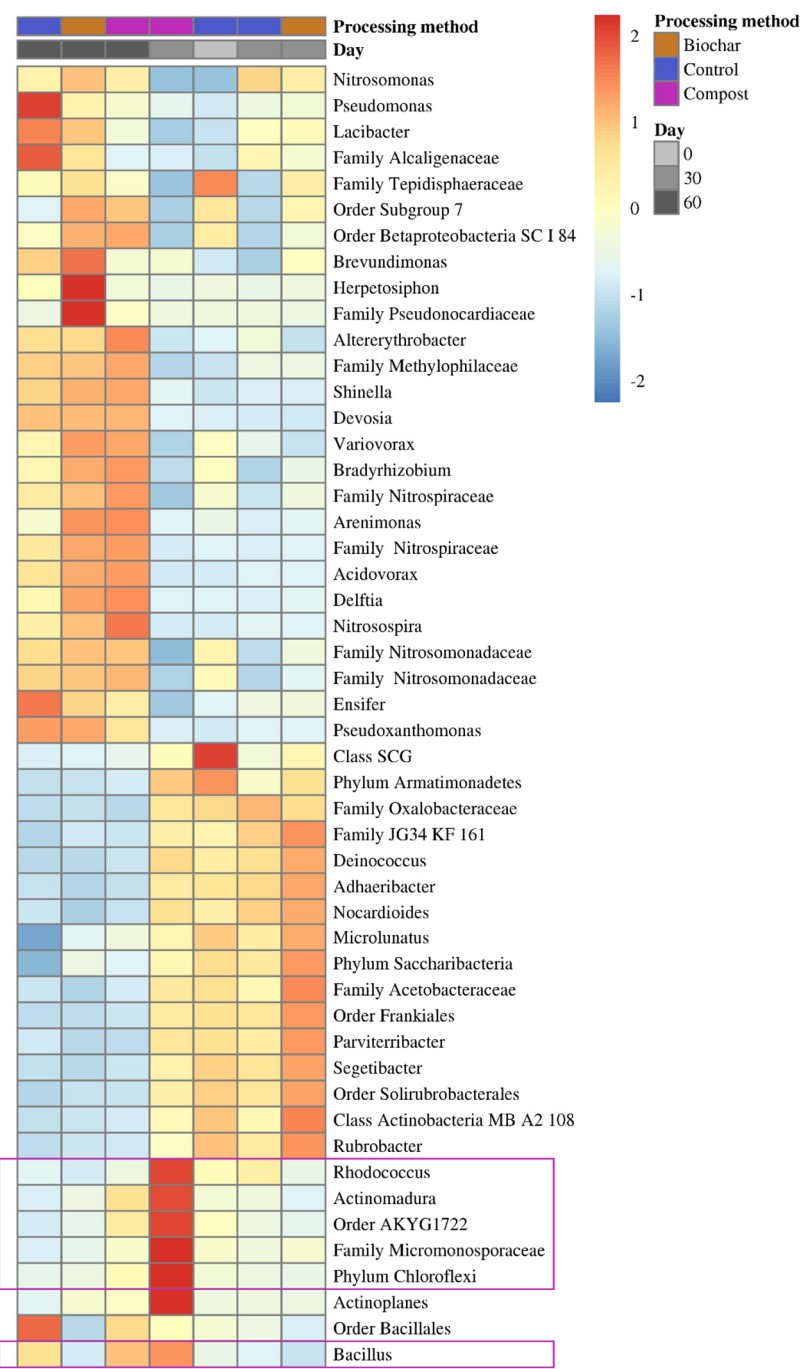

**Figure 8.** Bacterial taxonomic groups, which significantly differ in their abundances between processing methods and days, were identified using limma with additive log-ratio normalized counts. A *p*-value of <0.05 was considered as significant. The individual taxa are represented using a colour-code (center-scaled mean of read counts out of four replicates). The purple squares indicate taxa which are significantly different between processing methods without considering the days as covariates.

## 4. Discussion

We observed a significant change in the soil fungal and bacterial communities over the vegetation period of Chinese cabbage as well as significant changes in bacterial species richness, regardless of amendment type. The change could be attributed to changes in physicochemical parameters of soil during the experiment (pH, SOM content) and/or plant growth [54,55].

Fungal or bacterial species richness was not affected by amending the soil with SS. The obtained result is in line with our previous work [17], where no significant differences were observed regarding bacterial species richness when composted sewage sludge was applied to diluvial sand soil. Moreover, no difference was observed in fungal or bacterial species richness between the addition of compost and biochar. The fungal and bacterial community analysis showed that there was no difference between control soil from Day 0 and 30 and SS biochar amended soil on Day 30. Tayyab et al. [56] investigated the short-term impact of different organic amendments in soil, where significant influences of amendment type on fungal community structures were observed. Furthermore, fungal species richness decreased when manure containing amendments were added to soil [56]. On Day 60, the variability is higher in both fungal and bacterial communities compared to the previous timepoint. The impact of composted SS addition on soil fungal and bacterial communities was evident in both timepoints, while SS biochar amendment showed a shift only in the bacterial community on Day 60 compared to control soil.

One of the apparent phenomena influencing the microbial community in the rhizosphere is root exudation. Plants export about 30% of the assimilated $CO_2$ into the rhizosphere in the form of sugars, amino acids, phytosiderophores or carboxylates [57]. These compounds are freely used as nutrients by microorganisms, and therefore, the accumulation of exudates over the course of 60 days could partially explain the shift in the community composition. Curlango-Rivera et al. [58] investigated the effect of several cotton cultivars on the rhizosphere microbiome and concluded that the microbiome is shaped by the root exudates from the border cells depending on the observed cultivar promoting different microbiome compositions. Ali et al. [59] investigated different crop rotating systems, including Chinese cabbage and concluded that both the fungal and bacterial species were affected by the crop used, which in turn, altered the soil nutrient metabolism prolonging the intricate cycle at the soil-plant-microbiome interface and reducing the environmental risk of soil contaminants.

Furthermore, the change in soil microbiome is closely linked to changes in soil physicochemical properties rather than the introduction of new microbiota through soil amendments [54]. Stacey et al. [60] amended golf course soil with composted SS over a 2-year period and found that microbial communities were not strongly impacted by the amendment, but rather by soil edaphic factors. The shift in relative abundances of fungi with both amendments was more pronounced than the shift in the bacterial community. The addition of SS amendments also increased relative abundances of unknown soil tax, a suggesting a change in soil eukaryotic microorganisms. The observed differences in fungal phyla between SS amended soils and control soil were significant only in Zygomycota relative abundance on the phylum level and Pezizomycetes and Chytridiomycetes relative abundance on the class level. Pezyzomycetes were found only in composted SS amended soil and Chytridiomycetes were more abundant in biochar SS soils compared to control and composted SS amended soils.

Our results showed a significant increase in Firmicutes and Thermomicrobia at the bacterial phylum level and Bacilli at the class level when composted SS was applied to soil compared to SS biochar amendment and control soil suggesting a change induced by composted SS microbiota, as shown from the community analyses. In our previous paper [17], we observed an increase in relative abundance of Proteobacteria, Actinobacteria, Bacteriodetes, Chloroflexi, and Verrucomicrobia as well as a decrease in relative abundance of Firmicutes, Acidobacteria, Crenarchaeota, and Nitrospirae when composted sewage sludge was applied in DS soil. Bai et al. [61] observed an increase in Chloroflexi, Planctomycetes, and Firmicutes as well as a decrease in Proteobacteria when sewage sludge was applied to saline soils.

In bacterial relative abundances, we observed a significant increase in Firmicutes and Thermomicrobia at the phylum level and Bacilli on the class level when composted SS was applied to soil compared to SS biochar amendment and control soil. Our results are similar to the findings of Azeem et al. [62] who observed significant differences in the

soil microbiome structure in the wood compost amended soils, whereas in wood biochar amended soils communities were more similar to the control soil.

Using differential abundance analyses, we identified a number of taxa that respond mainly to the sampling day. In the case of fungal taxa, we found specific treatments with higher abundance, which might have affected the processing methods and days in the analysis. However, we could also observe that some bacterial taxa are significantly different between composted SS samples at all sampling points compared to other samples.

The results from our study indicate that the bacterial and fungal communities initially respond differently to the amendment type, but after only 60 days following the application, the amendment-caused change is less evident.

**Author Contributions:** Conceptualization, S.G.B. and D.B.; formal analysis, N.M.; funding acquisition, D.B.; investigation, N.M., I.P. (Igor Palčić), M.Č., S.G.B., I.P. (Igor Pasković), J.P., Z.U. and D.B.; methodology, N.M.; visualization, N.M. and J.S.; writing—original draft, N.M., J.S. and A.S.; writing—review and editing, J.S., A.S., I.P. (Igor Palčić), M.Č., S.G.B., I.P. (Igor Pasković), J.P., Z.U. and D.B. All authors have read and agreed to the published version of the manuscript.

**Funding:** The Croatian Science Foundation and The Environmental Protection and Energy Efficiency Fund are acknowledged for full financial support of the study (contract No. PKP-2016-06-9041). The work of J.S. and A.S. was supported by funds of the Federal Ministry of Food and Agriculture (BMEL) based on a decision of the Parliament of the Federal Republic of Germany via the Federal Office for Agriculture and Food (BLE), grant number 2819HS005.

**Institutional Review Board Statement:** Not applicable.

**Informed Consent Statement:** Not applicable.

**Data Availability Statement:** The data are available within this article.

**Conflicts of Interest:** The authors declare no conflict of interest.

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
