# Peer review of "Pyrolyzed or Composted Sewage Sludge Application Induces Short-Term Changes in the Terra Rossa Soil Bacterial and Fungal Communities"

_sustainability, doi:10.3390/su141811382_

Round 1
Reviewer 1 Report
The authors briefly described the effects of sewage sludge composting and sewage sludge biochar application on soil microbial communities. Although this study is well designed and interesting, the major modifications are needed to further clarify how this soil management practice affects microbial community composition.
The abstract needs to be rewritten because the background description in this section is too lengthy, while the experimental design and results description are too sample, especially lack of a summary of the article.
The title of the article focuses on the soil type is Terra Rossa soil, but it is only mentioned in the last paragraph of the entire introduction. Therefore, the authors need to focus on explaining why this soil type is used in the introduction. In addition, the description of the research progress of sewage sludge in the introduction is too simple, instead it focuses on the application of organic fertilizer and biochar.
The experimental design, data analysis, and results were described vaguely as follows:
L84-98 What are the initial physicochemical biological properties of sewage sludge, and after compositing and biochar production.
L99-111, What about fertilization and water management during crop planting.
L112-113, What are the initial physicochemical biological properties of this soil used in the experiment? Please provide the classification name of the soil according to World Reference Base for Soil Resources
L120-122, What are the sequences of these primers?
L147-148, What is the detailed counts or the least sequences?
L149-150, what codes or software that used to calculate the microbial alpha and beta diversities?
L132-150, the raw sequencing data should be deposited at the NCBI Sequence Read Archive database, and listed the accession number of PRJNAXXX.
L154, What are Figure1-6? Here, the method, software, and references used to visualize the data should be detailed to description.
L164-165 and L175-176, revised as “a total of 4013 fungal OTUs or a total of 7324 bacterial OTUs”.
L278-283 and L291-297, the detailed changes of microbial compositions among different treatments should be described in the text.
The discussion is too simplistic and does not dig deep into the underlying mechanisms by which different treatments affect microbial community changes.
For example, the authors introduce the soil physicochemical parameters is very important in determining microbial community assembly (l308-309), and further indicated the changes in the soil microbiome is also closely linked to the changes in soil physicochemical properties (L331-332). However, the authors do not describe how these treatments affect soil physicochemical properties in the Results section. Therefore, the authors should add the data of soil physicochemical properties during crop cultivation and analyze their relationships between microbial communities composition.
L354-359, The description of the discussion is not limited to a simple comparison with previous studies, it should be do an in –depth analysis of the underlying reasons.
Author Response
The authors briefly described the effects of sewage sludge composting and sewage sludge biochar application on soil microbial communities. Although this study is well designed and interesting, the major modifications are needed to further clarify how this soil management practice affects microbial community composition.
Thank you for your comments
The abstract needs to be rewritten because the background description in this section is too lengthy, while the experimental design and results description are too sample, especially lack of a summary of the article.
The background description was reduced while details of the experiment were added
The title of the article focuses on the soil type is Terra Rossa soil, but it is only mentioned in the last paragraph of the entire introduction. Therefore, the authors need to focus on explaining why this soil type is used in the introduction. In addition, the description of the research progress of sewage sludge in the introduction is too simple, instead it focuses on the application of organic fertilizer and biochar.
A paragraph has been added to Introduction which addresses the issue raised by the reviewer
The experimental design, data analysis, and results were described vaguely as follows:
L84-98 What are the initial physicochemical biological properties of sewage sludge, and after compositing and biochar production.
As reviewer requested, the data on initial physicochemical characteristics of sewage sludge, sludge compost and sludge biochar were provided (see the revised paragraph).
L99-111, What about fertilization and water management during crop planting.
On a reviewer's comment, the additional explanation was added (see the revised paragraph).
L112-113, What are the initial physicochemical biological properties of this soil used in the experiment? Please provide the classification name of the soil according to World Reference Base for Soil Resources
As reviewer requested, the terra rossa intitial characteristics and its classification was provided and added to the text (see the revised paragraph).
L120-122, What are the sequences of these primers?
References stating the sequences of the primers were added
L147-148, What is the detailed counts or the least sequences?
44481 for 16S and 45527 for ITS
L149-150, what codes or software that used to calculate the microbial alpha and beta diversities?
The MicrobiomeAnalystR package was employed (doi:10.1038/s41596-019-0264-1)
L132-150, the raw sequencing data should be deposited at the NCBI Sequence Read Archive database, and listed the accession number of PRJNAXXX.
The sequences are now deposited in the SRA under the number PRJNA871992 (see in the text)
L154, What are Figure1-6? Here, the method, software, and references used to visualize the data should be detailed to description.
The sentence was rephrased for better clarity
L164-165 and L175-176, revised as “a total of 4013 fungal OTUs or a total of 7324 bacterial OTUs”.
Revised accordingly
L278-283 and L291-297, the detailed changes of microbial compositions among different treatments should be described in the text.
Thank you for the comment. We added more precise information. Since it is difficult to describe the overall composition, we focused on the differential abundance of taxa between treatments or days. We believe that it might be a bit much to describe every single taxa.
The discussion is too simplistic and does not dig deep into the underlying mechanisms by which different treatments affect microbial community changes.
For example, the authors introduce the soil physicochemical parameters is very important in determining microbial community assembly (l308-309), and further indicated the changes in the soil microbiome is also closely linked to the changes in soil physicochemical properties (L331-332). However, the authors do not describe how these treatments affect soil physicochemical properties in the Results section. Therefore, the authors should add the data of soil physicochemical properties during crop cultivation and analyze their relationships between microbial communities composition.
Thank you for your comment. Unfortunately, we have only the required data for the last timepoint and as such would not show the entire story, so we omitted this statistical approach
L354-359, The description of the discussion is not limited to a simple comparison with previous studies, it should be do an in –depth analysis of the underlying reasons.
A paragraph was added with a description of the soil-plant-microbiome interaction to the Discussion section
Reviewer 2 Report
The manuscript presents a study of the short-term impact of sewage sludge in its composted or biochar form on the fungal and bacterial communities of the Terra Rossa soil by high-throughput sequencing. I think that the manuscript presents interesting results and the research objectives have been achieved. The study could provide helpful information to help enrich the knowledge on sewage sludge in different forms on the fungal and bacterial communities. However, I have concerns that should be addressed before the paper could be published.
1.In the Introduction part, the authors introduced the importance, the advantages and disadvantages of incorporation of organic fertilizers or other organic amendments into the soil. However, it seems insufficient for the latest researches on the effects of sewage sludge on the fungal and bacterial communities, especially the similar researches in other research regions or patterns. What’s more, the novelty in the study should be highlighted in a clearer way based on the systematic review of the related researches.
2. In the Materials and Methods part, please supplement the related information on the research region, climate types, soil types and cropping. The generation of the sewage sludge and the biochar should be also introduced to make the readers understand the manuscript in a clearer way. Moreover, the reasons why the dose of 40 kg of straw per 1m3 of sludge and the Chinese cabbage were adopted should be given more details. In addition, the authors mentioned “Amending soil with substrates was performed based on P fertilizing requirements according to the application rate of 118 mg P/L.”, it is better for the authors to supplement the related information on the reason why amending soil with substrates was performed based on P fertilizing requirements rather than other fertilizing requirements. In 3 of 20, line 112-113, the authors mentioned “The Terra Rossa soil used in substrate preparation was obtained locally and collected from the rhizosphere zone (0-30 cm depth) on the Istrian peninsula, Croatia.”, it is better for the authors to supplement the information on the reason why the 0-30 cm depth was selected as well as on the sampling method in regard of the rhizosphere zone.
3. In the Results part, the authors mentioned “The focus of this study is to investigate the effect of amending the Terra Rossa soil with SS with different processing methods on the soil fungal and bacterial communities. The experiment was carried out during a 60-day vegetation period of Chinese cabbage (Brassica rapa L.).” in the beginning of the part. It seems a little unsuitable for the contents to appear. It is better for the authors to introduce the results in a clear and concise way.
4. In the Results part, it seems insufficient for the authors to discuss the findings in this study. It is better for the authors to supplement and highlight the significance of the important findings, by such way of replying to requirements of environmentally sound processing methods of sewage sludge and the open and challenging issue for the European Union. Are the findings suitable to extend into the similar research region or types? Considering that the authors have mentioned the effects of the treatments and the sampling points, the importance and potential should be stressed for the possible application of SS management through its recycling in agriculture. It is better for the authors to give their answers. Moreover, it is better for the authors to supplement the possible role of the Chinese cabbage variety in affecting the fungal and bacterial communities of the Terra Rossa soil, although the same variety was selected in the trial.
5. And several faults, including format problems (Line 14 Soil increase Soil Organic Matter (SOM)), grammar errors (Line 60, On the other however; Line 78, reduce leeching of) occurred in this manuscript. It is better if the authors could check them carefully.
Author Response
The manuscript presents a study of the short-term impact of sewage sludge in its composted or biochar form on the fungal and bacterial communities of the Terra Rossa soil by high-throughput sequencing. I think that the manuscript presents interesting results and the research objectives have been achieved. The study could provide helpful information to help enrich the knowledge on sewage sludge in different forms on the fungal and bacterial communities. However, I have concerns that should be addressed before the paper could be published.
Thank you for your time and effort for reviewing our paper. Please find our answers below.
1.In the Introduction part, the authors introduced the importance, the advantages and disadvantages of incorporation of organic fertilizers or other organic amendments into the soil. However, it seems insufficient for the latest researches on the effects of sewage sludge on the fungal and bacterial communities, especially the similar researches in other research regions or patterns. What’s more, the novelty in the study should be highlighted in a clearer way based on the systematic review of the related researches.
Thank you for the comment. We are aware of the very dynamic development in this field of research. Since organic fertilizers or other organic amendments are being increasingly used in agriculture, quite many research groups focus on their impact on soil microbial communities. In our study, we did not intended to present a systematic review of the related research. Rather, we focused on the introduction of the most important aspects.
- In the Materials and Methods part,
please supplement the related information on the research region, climate types, soil types and cropping.
As reviewer requested, the additional information on the soil type, climate and cropping were provided (see the last paragraph of Materials and Methods).
The generation of the sewage sludge and the biochar should be also introduced to make the readers understand the manuscript in a clearer way.
The authors added additional infomation to the text (see revised section 2.1. Experimental setup).
Moreover, the reasons why the dose of 40 kg of straw per 1m3 of sludge and the Chinese cabbage were adopted should be given more details.
The authors added additional explanation on compost preparation to the text (see the revised section 2.1. Experimental setup).
Regarding Chinese cabbage, it was used as a model plant as part of a previous study concerning metal transfer from soil to Chinese cabbage due to its ability to accumulate potentially toxic elements.
In addition, the authors mentioned “Amending soil with substrates was performed based on P fertilizing requirements according to the application rate of 118 mg P/L.”, it is better for the authors to supplement the related information on the reason why amending soil with substrates was performed based on P fertilizing requirements rather than other fertilizing requirements.
The P-based application of SS compost or SS biochar was considerd since Terra Rossa is insufficiently supplied by plant-available P. The related explanation was added to the text (see the revised section 2.1. Experimental setup).
In 3 of 20, line 112-113, the authors mentioned “The Terra Rossa soil used in substrate preparation was obtained locally and collected from the rhizosphere zone (0-30 cm depth) on the Istrian peninsula, Croatia.”, it is better for the authors to supplement the information on the reason why the 0-30 cm depth was selected as well as on the sampling method in regard of the rhizosphere zone.
The authors highlighted and explained, both the reason on sampling in the rhizosphere, as well as the sampling method used (see the revised section 2.1. Experimental setup).
- In the Results part, the authors mentioned “The focus of this study is to investigate the effect of amending the Terra Rossa soil with SS with different processing methods on the soil fungal and bacterial communities. The experiment was carried out during a 60-day vegetation period of Chinese cabbage (Brassica rapa L.).” in the beginning of the part. It seems a little unsuitable for the contents to appear. It is better for the authors to introduce the results in a clear and concise way.
The sentence was removed
- In the Results part, it seems insufficient for the authors to discuss the findings in this study. It is better for the authors to supplement and highlight the significance of the important findings, by such way of replying to requirements of environmentally sound processing methods of sewage sludge and the open and challenging issue for the European Union. Are the findings suitable to extend into the similar research region or types? Considering that the authors have mentioned the effects of the treatments and the sampling points, the importance and potential should be stressed for the possible application of SS management through its recycling in agriculture. It is better for the authors to give their answers.
Thank you for your comment. This experiment is a step forward, but more extensive research is needed to fully understand the effect of SS management through its application in soil and to give definitive answers on this matter.
Moreover, it is better for the authors to supplement the possible role of the Chinese cabbage variety in affecting the fungal and bacterial communities of the Terra Rossa soil, although the same variety was selected in the trial.
A paragraph was added to the discussion section concerning the effect of the plant on the soil microbiome
- And several faults, including format problems (Line 14 Soil increase Soil Organic Matter (SOM)), grammar errors (Line 60, On the other however; Line 78, reduce leeching of) occurred in this manuscript. It is better if the authors could check them carefully.
Done
Reviewer 3 Report
The aim of the paper “Pyrolyzed or composted sewage sludge application induces short-term changes in the Terra Rossa soil bacterial and fungal communities” was to evaluate the short-term impact of sewage sludge in its composted or biochar form on the fungal and bacterial communities of the Terra Rossa soil. Sewage sludge is a challenging issue for many countries. It is interesting and important to know how to apply these wastes in a reasonable and safe way. This article falls within the scope of this journal, and I believe that it will be interesting to readers.
The experiment focused in two different processing methods for sewage sludge. However, the authors did not do any tests on the sludge itself, on the compost and biochar made from the sludge, or on the soil used for pot trail. Sequencing of soil with amendment of compost and biochar described changes in its microbial community, but could not explain the reasons for these changes. The basic physicochemical properties, nutrient information of these substrates are very important to explain the changes in soil microbial communities.
The authors should perform a detailed analysis of sewage sludge, compost, biochar, and soil for potting, and perform a multivariate analysis of these results with bacterial and fungal sequencing data to explore the reasons for the impact of compost application on the soil microbial community, which will tell a full story.
Line 31-81: Paragraph formatting should be set to align at both ends.
Line 39: “soil quality is a complex matter involving … time”, this is a confusing expression, do the authors mean that time is a soil quality factor?
Line 73: sewage sludge first appear in line 70, the abbreviations should be there.
I suggest the authors provide more insight into how different organic amendments affect soil fertility and the impact of soil microbial diversity in the introduction.
Line 87: The start of the composting experiment was listed, but what about the duration of the composting? To what extent does composting stop?
Line 95: The compost maturity and its quality mentioned here should be listed in the result.
Line 104: The amount of soil, SS compost and biochar should be clearly stated.
Line 108-110: What is the nutrient content of the compost and biochar? On what standard did the authors applying the P amendment?
Line 112-113: Did the authors do mixing and sieving to homogenize the soil?
The description of soil and substrate preparation should be placed before the description of the pot trail.
Author Response
The aim of the paper “Pyrolyzed or composted sewage sludge application induces short-term changes in the Terra Rossa soil bacterial and fungal communities” was to evaluate the short-term impact of sewage sludge in its composted or biochar form on the fungal and bacterial communities of the Terra Rossa soil. Sewage sludge is a challenging issue for many countries. It is interesting and important to know how to apply these wastes in a reasonable and safe way. This article falls within the scope of this journal, and I believe that it will be interesting to readers.
Thank you for your comments
The experiment focused in two different processing methods for sewage sludge. However, the authors did not do any tests on the sludge itself, on the compost and biochar made from the sludge, or on the soil used for pot trail. Sequencing of soil with amendment of compost and biochar described changes in its microbial community, but could not explain the reasons for these changes. The basic physicochemical properties, nutrient information of these substrates are very important to explain the changes in soil microbial communities.
Thank you for your comment. The physicochemical characteristics of the materials were added to the M&M section
The authors should perform a detailed analysis of sewage sludge, compost, biochar, and soil for potting, and perform a multivariate analysis of these results with bacterial and fungal sequencing data to explore the reasons for the impact of compost application on the soil microbial community, which will tell a full story.
Thank you for your comment. Unfortunately, we have only the required data for the last timepoint and as such would not show the entire story, so we omitted this statistical approach
Line 31-81: Paragraph formatting should be set to align at both ends.
Done
Line 39: “soil quality is a complex matter involving … time”, this is a confusing expression, do the authors mean that time is a soil quality factor?
The sentence was rephrased for better clarity
Line 73: sewage sludge first appear in line 70, the abbreviations should be there.
Done
I suggest the authors provide more insight into how different organic amendments affect soil fertility and the impact of soil microbial diversity in the introduction.
More information was provided about the amendment impact on soil fertility and soil microbial diversity (see the revised Introduction).
Line 87: The start of the composting experiment was listed, but what about the duration of the composting? To what extent does composting stop?
The composting process lasts for about three months. The additional explanation was added to the text with the supporting references (see the revised section 2.1. Experimental setup).
Line 95: The compost maturity and its quality mentioned here should be listed in the result.
The physicochemical properties of compost and biochar were provided by adding the relevant reference (Černe et al., 2022) to 2.1. Experimental setup section where the initial characteristics of raw materials were described.
Reference: Černe, M., Palčić, I., Major, N., Pasković, I., Perković, J., Užila, Z., Filipović, V., Romić, M., Goreta Ban, S., Heath, D.J., & Ban, D. (2022). Effect of sewage sludge-derived amendments on the nutrient uptake by Chinese cabbage from Mediterranean soils. Journal of Plant Nutrition, https://doi.org/10.1080/01904167.2022.2071732
Line 104: The amount of soil, SS compost and biochar should be clearly stated.
The amounts of soil, compost and biochar were provided for individual pot (3 L) and indicated in section 2.1. Experimental setup.
Line 108-110: What is the nutrient content of the compost and biochar? On what standard did the authors applying the P amendment?
The nitrogen, phosphorous and potassium concentrations (mentioned as intitial physicochemical characteristics) were provided for compost and biochar in section 2.1. Experimental setup by citing the relevant reference (Černe et al., 2022). Also, the P-based application was performed according to P fertilising demands for Chinese cabbage growing (see the revised section 2.1. Experimental setup).
Line 112-113: Did the authors do mixing and sieving to homogenize the soil?
The soil was mixed but it was not sieved since it was used in its undisturbed form, but the authors do sieved (2 mm mesh) the compost and biochar samples to attain their homogenous distribution in soil. All these is already explained in section 2.1. Experimental setup.
The description of soil and substrate preparation should be placed before the description of the pot trail.
Done
Round 2
Reviewer 1 Report
Well done.
-
The author has made effective modifications and answers to the questions raised before.
Author Response
Thank you for your comments!
Reviewer 3 Report
The details of Figure 1, 2, 3 and 4 (color, borders, size, etc.) can be further optimized.
line 147, an extra period.
Author Response
Thank you for your comments.
The frame was removed from the Figures and the font changed to suit the Journal.
The extra dot was removed.